# Uncertainty-based Continual Learning with Adaptive Regularization

**Hongjoon Ahn**[1]*, **Sungmin Cha**[2]*, **Donggyu Lee**[2] **and Taesup Moon**[1,2]
[1] Department of Artificial Intelligence, [2]Department of Electrical and Computer Engineering,
Sungkyunkwan University, Suwon, Korea 16419
{hong0805, csm9493, ldk308, tsmoon}@skku.edu

## Abstract

We introduce a new neural network-based continual learning algorithm, dubbed as Uncertainty-regularized Continual Learning (UCL), which builds on traditional Bayesian online learning framework with variational inference. We focus on two significant drawbacks of the recently proposed regularization-based methods: a) considerable additional memory cost for determining the per-weight regularization strengths and b) the absence of gracefully forgetting scheme, which can prevent performance degradation in learning new tasks. In this paper, we show UCL can solve these two problems by introducing a fresh interpretation on the Kullback-Leibler (KL) divergence term of the variational lower bound for Gaussian mean-field approximation. Based on the interpretation, we propose the notion of node-wise uncertainty, which drastically reduces the number of additional parameters for implementing per-weight regularization. Moreover, we devise two additional regularization terms that enforce *stability* by freezing important parameters for past tasks and allow *plasticity* by controlling the actively learning parameters for a new task. Through extensive experiments, we show UCL convincingly outperforms most of recent state-of-the-art baselines not only on popular supervised learning benchmarks, but also on challenging lifelong reinforcement learning tasks. The source code of our algorithm is available at https://github.com/csm9493/UCL.

## 1 Introduction

Continual learning, also called as lifelong learning, is a long-standing open problem in machine learning in which data from multiple tasks continuously arrive and the learning algorithm should constantly adapt to new tasks as well as not forget what it has learned in the past. The main challenge is to resolve the so-called *stability-plasticity dilemma* [2, 18]. Namely, a learning agent should be able to preserve what it has learned, but focusing too much on the stability may hinder it from quickly learning a new task. On the other hand, when the agent focuses too much on the plasticity, it tends to quickly forget what it has learned. Particularly, for the artificial neural network (ANN)-based models, which became the mainstream of the machine learning methods, it is well-known that they are prone to such *catastrophic forgetting* phenomenon [17, 4]. As opposed to the ANNs, humans are able to maintain the obtained knowledge while learning a new task, and the forgetting in human brain happens gradually rather than drastically. This difference motivates active research in developing neural network based continual learning algorithms.

As given in a comprehensive survey [20] on this topic, approaches for tackling the catastrophic forgetting in neural network based continual learning can be roughly grouped into three categories: regularization-based [14, 12, 30, 19], dynamic network architecture-based [23, 29], and dual memory system-based [22, 15, 27, 10]. While each category has its own merit, of particular interest are the

---

regularization-based methods, since they pursue to maximally utilize the limited network capacity by imposing constraints on the update of the network given a new task. Computationally, they typically are realized by adding regularization terms that penalize the changes in the network parameters when learning a new task. This approach makes sense since it is well-known that neural network models are highly over-parametrized, and once successful, it can be also complementary to other approaches since it can lead to the efficient usage of network capacity as the number of tasks grows, as in [25].

The recent state-of-the-art regularization-based methods typically implement the per-parameter regularization parameters based on several different principles inferring the importance of each parameter for given tasks; e.g., diagonal Fisher information matrix for EWC [12], variance term associated with each weight parameter for VCL [19], and the path integrals of the gradient vector fields for SI [30]. While these methods are shown to be very effective in several continual learning benchmarks, a common caveat is that the amount of the memory required to store the model is twice the original neural network parameters, since they need to store the individual regularization parameters. We note that this could be a limiting factor for being deployed with large network size.

In this paper, we propose a new regularization-based continual learning algorithm, dubbed as Uncertainty-regularized Continual Learning (UCL), that stores much smaller number of additional parameters for regularization terms than the recent state-of-the-arts, but achieves much better performance in several benchmark datasets. Followings summarize our key contributions.

- We adopt the standard Bayesian online learning framework, but make a fresh interpretation of the Kullback-Leibler (KL) divergence term of the variational lower bound for the Gaussian mean-field approximation case.
- We define a novel notion of "uncertainty" for each hidden node in a network by tying the learnable variances of the incoming weights of a node. Moreover, we add two additional regularization terms to *freeze* the weights that are identified to be important and to *gracefully forget* what was learned before and control the actively learning weights.
- We achieve state-of-the-art performances on a number of continual learning benchmarks, including supervised learning (SL) tasks with deep convolutional neural networks and reinforcement learning (RL) tasks with different state-action spaces. Performing well on both SL and RL continual learning tasks is a unique strength of our UCL.

## 2   Related Work

**Continual learning**   There are numerous approaches in continual learning and we refer the readers to [20] for an extensive review. We only list work relevant to our method. The main approach of regularization-based methods in continual learning is to identify the important weights for the learned tasks and penalize the large updates on those weights when learning a new task. LwF [14] contains task-specific layers, and keeps the similar outputs for the old tasks by utilizing knowledge distillation [9]. In EWC [12], the diagonal of the Fisher information matrix at the learned parameter of the given task is used for giving the relative regularization strength. An extended version of EWC, IMM [13], merged the posteriors based on the mean and the mode of the old and new parameters. SI [30] computes the parameter importance considering a path integral of gradient vector fields during the parameter updates. VCL [19] also adopts Bayesian online learning framework as ours, but simply applies standard techniques that results in some drawbacks, which are elaborated in Section 3.1.

Some work approached continual learning differently than the regularization-based method for the limited network capacity case. PackNet [16] picks out task-specific weights based on the weight pruning method, which requires saving the learnable binary masks for the weights. HAT [26] employs node-wise attention mechanism per layer using the task identifier embedding, but requires a knowledge on the number of tasks *a priori*, which is a critical limitation.

**Variational inference**   In standard Bayesian learning, the main idea of learning is efficiently approximating the posterior distribution on the models. [6] introduces a practical variational inference technique for neural networks, which suggested that variational parameters can be learned using back-propagation. Another approach in variational inference is [11] which introduces the approximated lower bound of likelihood, and learn variational parameter using re-parameterization trick. In [1], they introduce Unbiased Monte Carlo, which also uses back-propagation, but many kinds of priors can be used in the Unbiased Monte Carlo. In addition, there are several practical methods for variational inference in neural networks, such as using dropout [5] or Expectation-Propagation [8].

# 3 Uncertainty-regularized Continual Learning (UCL)

## 3.1 Notations and a review on Bayesian online learning

Consider a discriminative neural network model, $p(\boldsymbol{y}|\mathbf{x},\mathcal{W})$, that returns a probability distribution over the output $\boldsymbol{y}$ given an input $\mathbf{x}$ and parameters $\mathcal{W}$. In standard Bayesian learning, $\mathcal{W}$ is assumed to be sampled from some prior distribution $p(\mathcal{W}|\boldsymbol{\alpha})$ that depends on some parameter $\boldsymbol{\alpha}$, and after observing some data $\mathcal{D} = \{(\boldsymbol{x}_i, y_i)\}_{i=1}^n$, obtaining the posterior $p(\mathcal{W}|\boldsymbol{\alpha}, \mathcal{D})$ becomes the central problem to learn the model parameters. Since exactly obtaining the posterior becomes intractable, variational inference [1, 3, 6] instead tries to approximate this posterior with a more tractable distribution $q(\mathcal{W}|\boldsymbol{\theta})$. The approximation is done by minimizing (over $\boldsymbol{\theta}$) the so-called *variational free energy*, which can be written as

$$\mathcal{F}(D, \boldsymbol{\theta}) = \mathbb{E}_{q(\mathcal{W}|\boldsymbol{\theta})}[-\log p(D|\mathcal{W})] + D_{KL}(q(\mathcal{W}|\boldsymbol{\theta})\|p(\mathcal{W}|\boldsymbol{\alpha})), \tag{1}$$

in which $\log p(D|\mathcal{W})$ is the log-likelihood of the data $D$ determined by the model $p(\boldsymbol{y}|\mathbf{x},\mathcal{W})$, and $D_{KL}(\cdot)$ is the Kullback-Leibler divergence. Moreover, the commonly used $q(\mathcal{W}|\boldsymbol{\theta})$ is the so-called Gaussian mean-field approximation, $q(\mathcal{W}|\boldsymbol{\theta}) = \prod_i \mathcal{N}(w_i|\mu_i, \sigma_i)$ with $\boldsymbol{\theta} = (\boldsymbol{\mu}, \boldsymbol{\sigma})$, and $\boldsymbol{\theta}$ can be learned via reparameterization trick [11] and the standard back-propagation.

In Bayesian online learning framework, standard variational inference can be applied to the continual learning setting. Namely, when a dataset for task $t$, $\mathcal{D}_t$ arrives, the framework solves to minimize

$$\mathcal{F}(D_t, \boldsymbol{\theta}_t) = \mathbb{E}_{q(\mathcal{W}|\boldsymbol{\theta}_t)}[-\log p(D_t|\mathcal{W})] + D_{KL}(q(\mathcal{W}|\boldsymbol{\theta}_t)\|q(\mathcal{W}|\boldsymbol{\theta}_{t-1})) \tag{2}$$

over $\boldsymbol{\theta}_t = (\boldsymbol{\mu}_t, \boldsymbol{\sigma}_t)$, in which $q(\mathcal{W}|\boldsymbol{\theta}_{t-1})$ stands for the posterior learned after observing $\mathcal{D}_{t-1}$ acting as a prior for learning $q(\mathcal{W}|\boldsymbol{\theta}_t)$. Note in (2), we can observe that the KL-divergence term naturally acts as a regularization term. In VCL [19], they showed that the network learned by sequentially solving (2) by using projection operator of the variational inference for each task $t$ can successfully combat the catastrophic forgetting problem to some extent.

However, we argue that this Bayesian approach of VCL has several drawbacks as well. First, due to the Monte-Carlo sampling of the model weights for computing the likelihood term in (2), the time and space complexity for learning grows with the sample size. Second, since the variance term is defined for every weight parameter, the number of parameters to maintain becomes exactly twice the size of network weights. This becomes problematic when deploying a large-sized network, as is the case in modern deep learning. In this paper, we present a novel approach which can resolve above problems. Our key idea is rooted in a fresh interpretation of the closed form of KL-divergence term in (2) for the Gaussian mean-field approximation and the Bayesian neural network pruning [6, 1].

## 3.2 Interpreting KL-divergence and motivation of UCL

While the KL divergence in (2) acts as a generic regularization term, we give a closer look at it, particularly for the Gaussian mean-field approximation case. Namely, after some algebra and evaluating the Gaussian integral, the closed-form of $D_{KL}(q(\mathcal{W}|\boldsymbol{\theta}_t)\|q(\mathcal{W}|\boldsymbol{\theta}_{t-1}))$ becomes:

$$\frac{1}{2}\sum_{l=1}^{L}\Big[\underbrace{\Big\|\frac{\boldsymbol{\mu}_t^{(l)} - \boldsymbol{\mu}_{t-1}^{(l)}}{\boldsymbol{\sigma}_{t-1}^{(l)}}\Big\|_2^2}_{(a)} + \underbrace{\mathbf{1}^\top\Big\{\Big(\frac{\boldsymbol{\sigma}_t^{(l)}}{\boldsymbol{\sigma}_{t-1}^{(l)}}\Big)^2 - \log\Big(\frac{\boldsymbol{\sigma}_t^{(l)}}{\boldsymbol{\sigma}_{t-1}^{(l)}}\Big)^2\Big\}}_{(b)}\Big], \tag{3}$$

in which $L$ is the number of layers in the network, $(\boldsymbol{\mu}_t^{(l)}, \boldsymbol{\sigma}_t^{(l)})$ are the mean and standard deviation of the weight matrix for layer $l$ that are subject to learning for task $t$, $(\boldsymbol{\mu}_{t-1}^{(l)}, \boldsymbol{\sigma}_{t-1}^{(l)})$ are the same quantity that are learned up to the previous task, the fraction notation means the element-wise division between tensors, and $\|\cdot\|_2^2$ stands for the Frobenius norm of a matrix. The detailed derivation of (3) is given in the Supplementary Materials. the term (a) in (3) can be interpreted as a square of the Mahalanobis distance between the vectorized $\boldsymbol{\mu}_t^{(l)}$ and $\boldsymbol{\mu}_{t-1}^{(l)}$, in which the covariance matrix is $\mathbf{diag}((\boldsymbol{\sigma}_{t-1}^{(l)})^2)$, and it acts as a regularization term for $\boldsymbol{\mu}_t^{(l)}$. Namely, when minimizing (3) over $\boldsymbol{\theta}_t^{(l)} = (\boldsymbol{\mu}_t^{(l)}, \boldsymbol{\sigma}_t^{(l)})$, the inverse of the variance learned up to task $(t-1)$ is acting as per-weight regularization strengths for $\boldsymbol{\mu}_t^{(l)}$ deviating from $\boldsymbol{\mu}_{t-1}^{(l)}$. This makes sense since each element of $(\boldsymbol{\sigma}_{t-1}^{(l)})^2$ can be regarded as an *uncertainty* measure for the corresponding mean weight of $\boldsymbol{\mu}_{t-1}^{(l)}$, and a weight with small uncertainty

should be treated as *important* such that high penalty is imposed when significantly getting updated for a new task $t$. Moreover, the term (b) in (3), which is convex in $(\sigma_t^{(l)})^2$ and is minimized when $\sigma_t^{(l)} = \sigma_{t-1}^{(l)}$, is acting as a regularization term for $\sigma_t^2$. Note it promotes to preserve the learned uncertainty measure when updating for a new task. This also makes sense for preventing catastrophic forgetting since the weights identified as important in previous tasks should be kept as important for future tasks as well such that the weights do not get updated too much by the term (a). Based on this interpretation, we modify each term and devise a new loss function for UCL.

### 3.3 Modifying the term (a)

We modify the term (a) in (3) based on the following three intuitions. First, instead of maintaining the uncertainty measure for each mean weight parameter of $\boldsymbol{\mu}_t$, we devise a notion of uncertainty for each *node* of the network. Second, based on the node uncertainty, we set the high regularization strength for a weight when either of the nodes it connects has low uncertainty. Third, we add additional $\ell_1$-regularizer such that a weight gets even more stringent penalty for getting updated when the weight has large magnitude or low uncertainty, inspired by [1, 6]. We elaborate each of these intuitions below.

While it is plausible to maintain the weight-level importance as in other work [12, 19, 30], we believe maintaining the importance (or uncertainty in our case) at the level of node makes more sense, not only for the purpose of reducing the model parameters, but also because the node value (or the activation) is the basic unit for representing the learned information from a task. A similar intuition of working at node-level also appears in HAT [26], which devised a hard attention mechanism for important *nodes*, or dropout [28], which randomly drops *nodes* while training. In our setting, we define the uncertainty of a node as illustrated in Figure 1; first constrain the incoming weights to the node to have the *same* standard deviation parameters as in node $j$ of layer $(l-1)$ in Figure 1, then set the variance as the uncertainty of the node. For the Gaussian mean-field approximation case, this constraint corresponds to adding zero-mean i.i.d Gaussian noise (with difference variances for different nodes) to the incoming weights when sampling for the variational learning.

For our second intuition, we derive the weight-level regularization scheme based on the following arguments. Namely, as shown in Figure 1, suppose a node is identified as important (the orange nodes), *i.e.*, has low uncertainty, for the past tasks, and the learning of a new task is taking place. We believe there are two major sources that can cause the catastrophic forgetting of the past tasks when a weight update for a new task happens; 1) the negative transfer (blue region) happening in the incoming weights of an important node, and 2) the information loss (pink region) happening in the outgoing weights of an important node. From the perspective of the important node, it is clear that when any of the incoming weights are significantly updated during the learning of the new task, the node's representation of the past tasks will significantly get altered as the node will differently combine information from the lower layer, and hurt the past tasks accuracy.

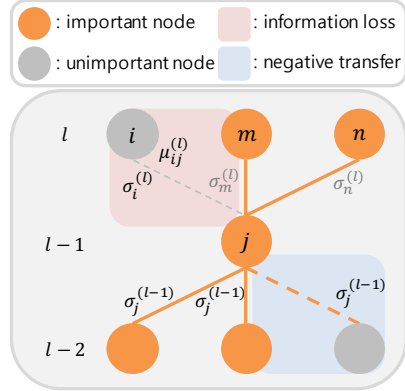

Figure 1: Information loss and negative transfer of an important node.

On the other hand, when the outgoing weights of the important node are significantly updated, the information of that node will get washed out during forward propagation, hence, it may not play an important role in computing the prediction, causing the accuracy drop for the past task.

From above argument, we devise the weight-level regularization such that weight gets high regularization strength when either of the node it connects has low uncertainty. This is realized by replacing the term (a) of (3) with the following:

$$\frac{1}{2}\Big(\sum_{l=1}^{L}\Big\|\boldsymbol{\Lambda}^{(l)}\odot(\boldsymbol{\mu}_t^{(l)}-\boldsymbol{\mu}_{t-1}^{(l)})\Big\|_2^2\Big), \quad \text{where} \quad \boldsymbol{\Lambda}_{ij}^{(l)}\triangleq\max\Big\{\frac{\sigma_{\text{init}}^{(l)}}{\sigma_{t-1,i}^{(l)}},\frac{\sigma_{\text{init}}^{(l-1)}}{\sigma_{t-1,j}^{(l-1)}}\Big\}, \tag{4}$$

in which $\sigma_{\text{init}}^{(l)}$ is the initial standard deviation hyperparameter for all weights on the $l$-th layer, $L$ is the number of layers in the network, $\boldsymbol{\mu}_t^{(l)}$ is the mean weight matrix for layer $l$ and task $t$, $\odot$ is the element-wise multiplication between matrices, and the matrix $\boldsymbol{\Lambda}^{(l)}$ defines the regularization strength

for the weight $\mu_{t,ij}^{(l)}$; *i.e.*, when either $\sigma_{t-1,i}^{(l)}$ or $\sigma_{t-1,j}^{(l-1)}$ is small, $\mu_{t,ij}^{(l)}$ gets high regularization strength. We note setting $\sigma_{\text{init}}^{(l)}$ correctly is important to control the stability of the learning process.

While (4) is a sensible replacement of the term (a) in (3), our third intuition above is based on the observation that (4) does not take into account of the magnitude of the learned weights, i.e., $\boldsymbol{\mu}_{t-1}^{(l)}$. In [1, 6], they applied a heuristic for *pruning* network weights learned by variational inference; *i.e.*, only keeps the weight if the magnitude of the ratio $\mu/\sigma$ is large, and prunes otherwise. Inspired by the pruning heuristic, we devise an additional $\ell_1$-norm based regularizer

$$\sum_{l=1}^{L}(\sigma_{\text{init}}^{(l)})^2\left\|\left(\frac{\boldsymbol{\mu}_{t-1}^{(l)}}{\boldsymbol{\sigma}_{t-1}^{(l)}}\right)^2\odot(\boldsymbol{\mu}_t^{(l)}-\boldsymbol{\mu}_{t-1}^{(l)})\right\|_1,\tag{5}$$

in which the division and square inside the $\ell_1$-norm should be understood as the element-wise operations. Note $\boldsymbol{\sigma}_{t-1}^{(l)}$ has the same dimension as $\boldsymbol{\mu}_{t-1}^{(l)}$, and the $i$-th row of $\boldsymbol{\sigma}_{t-1}^{(l)}$ has the same variance value associated with the $i$-th node in layer $l$. Thus, in (5), if the ratio $(\mu_{t-1,ij}^{(l)}/\sigma_{t-1,i}^{(l)})^2$ is large, the $\ell_1$-norm will promote sparsity and $\mu_{t,ij}^{(l)}$ will tend to *freeze* to $\mu_{t-1,ij}^{(l)}$.

### 3.4 Modifying the term (b)

Regarding the term (b) in (3), we can also devise a similar loss on the uncertainties associated with nodes. As mentioned in Section 3.2, the loss will promote $\boldsymbol{\sigma}_t^{(l)}=\boldsymbol{\sigma}_{t-1}^{(l)}$, meaning that once a node becomes important at task $(t-1)$, it tends to stay important for a new task as well. While this makes sense for preventing the catastrophic forgetting as it may induce high regularization parameters for penalties in (4) and (5), one caveat is that the network capacity can quickly fill up when the number of tasks grows. Therefore, we choose to add one more regularization term to the term (b) in (3),

$$\frac{1}{2}\mathbf{1}^\top\left((\boldsymbol{\sigma}_t^{(l)})^2-\log(\boldsymbol{\sigma}_t^{(l)})^2\right),\tag{6}$$

which inflates $\boldsymbol{\sigma}_t^{(l)}$ to get close to $\sqrt{2}\boldsymbol{\sigma}_{t-1}^{(l)}$ when minimized together with the term (b). The detailed derivation of the minimizer is given in the Supplementary Materials. Therefore, if a node becomes uncertain when training current task, the regularization strength becomes smaller. Since our initial standard deviation $\boldsymbol{\sigma}_{\text{init}}^{(l)}$ is usually set to be small, the additional term in (6) compared to the term (b) in (3) will tend to increase the number of "actively" learning nodes that have incoming weights with sufficiently large standard deviation values for *exploration*. Moreover, when a new task arrives while most of the nodes have low uncertainty, (6) will force some of them to increase the uncertainty level to learn the new task, resulting in *gracefully* forgetting the past tasks.

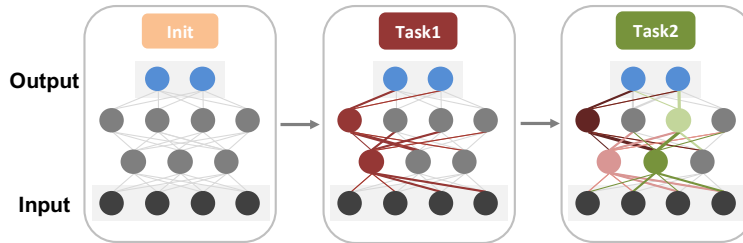

Figure 2: Colored hidden nodes and edges denote important nodes and highly regularized weights due to (4), respectively. The width of colored edge denotes the regularization strength of (5). Note as new task comes the uncertainty level of a node can vary due to (6), respresented with color changes.

### 3.5 Final loss function for UCL

Combining (4), (5), and (6), the final loss function for our UCL for task $t$ becomes

$$-\log p(D_t|\mathcal{W})+\sum_{l=1}^{L}\left[\left(\frac{1}{2}\left\|\boldsymbol{\Lambda}^{(l)}\odot(\boldsymbol{\mu}_t^{(l)}-\boldsymbol{\mu}_{t-1}^{(l)})\right\|_2^2+(\sigma_{\text{init}}^{(l)})^2\left\|\left(\frac{\boldsymbol{\mu}_{t-1}^{(l)}}{\boldsymbol{\sigma}_{t-1}^{(l)}}\right)^2\odot(\boldsymbol{\mu}_t^{(l)}-\boldsymbol{\mu}_{t-1}^{(l)})\right\|_1\right)\right.$$
$$\left.+\frac{\beta}{2}\mathbf{1}^\top\left\{\left(\frac{\boldsymbol{\sigma}_t^{(l)}}{\boldsymbol{\sigma}_{t-1}^{(l)}}\right)^2-\log\left(\frac{\boldsymbol{\sigma}_t^{(l)}}{\boldsymbol{\sigma}_{t-1}^{(l)}}\right)^2+(\boldsymbol{\sigma}_t^{(l)})^2-\log(\boldsymbol{\sigma}_t^{(l)})^2\right\}\right],\tag{7}$$

which is minimized over $\{\boldsymbol{\mu}_t^{(l)}, \boldsymbol{\sigma}_t^{(l)}\}_{l=1}^L$ and has two hyperparameters, $\{\sigma_{\text{init}}^{(l)}\}_{l=1}^L$ and $\beta$. The former serves as pivot values determining the degree of uncertainty of each node, and the latter controls the increasing or decreasing speed of $\boldsymbol{\sigma}_t^{(l)}$. As elaborated in above sections, it is clear that the uncertainty of a node plays a critical role in setting the regularization parameters, hence, justifies the name UCL. Illustration of the regularization mechanism of UCL is given in Figure 2. At the beginning epoch of task $t$, we sample from $q(\mathcal{W}|\boldsymbol{\theta}_t)$ with $\boldsymbol{\theta}_t = \boldsymbol{\theta}_{t-1}$, then continue to update $\boldsymbol{\theta}_t$ in the subsequent iterations. The model parameters are sampled every iteration, like in the usual Monte Carlo sampling, but we set the number of sampling to 1 for each iteration. This is an important differentiation that enables the application of UCL to reinforcement learning tasks, which was impossible for VCL [19].

## 4 Experimental Results

### 4.1 Supervised learning

We evaluate the performance of UCL together with EWC [12], SI [30], VCL [19], and HAT [26]. We also make a comparison with Coreset VCL proposed in [19]. The number of sampling weights was 10 for VCL, and 1 for UCL. All of the results are averaged over 5 different seeds. For the experiments with MNIST datasets, we used fully-connected neural networks (FNN), and with CIFAR-10/100 and Omniglot datasets, we used convolutional neural networks (CNN). The detailed architectures are given in each experiment section. Moreover, the initial standard deviations for UCL, $\{\sigma_{\text{init}}^{(l)}\}_{l=1}^L$, were set to be 0.06 for FNNs and adaptively set like the *He initialization* [7] for deeper CNNs, of which details are given in the Supplementary Material. The hyperparameter selections among the baselines are done fairly, and we list the selected hyperparameters in the Supplementary Materials.

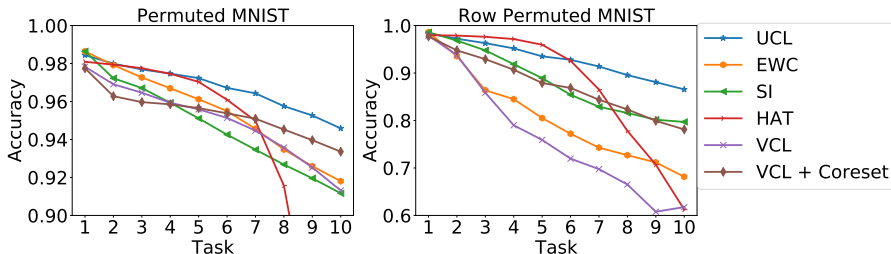

Figure 3: Experimental results on Permuted / Row Permuted MNIST with a single headed network.

**Permuted / Row Permuted MNIST** We first test on the popular Permuted MNIST dataset. We used single-headed FNN that has two hidden layers with 400 nodes and ReLU activations for all methods. We compare the average test accuracy over the learned tasks in Figure 3 (left). After training on 10 tasks sequentially, EWC, SI, and VCL show little difference of performance among them achieving 91.8%, 91.1%, and 91.3% respectively. Although VCL with the coreset size of 200 makes an improvement of 2%, UCL outperforms all other baselines achieving 94.5%. Interestingly, HAT keeps almost the same average accuracy as UCL until the first 5 tasks, but it starts to significantly deteriorate after task 7. This points out the limitation of applying HAT in a single-headed network. As a variation of Permuted MNIST, we shuffled only rows of MNIST images instead of shuffling all the image pixels, and we denoted it as Row Permuted MNIST. We empirically find that all algorithms are more prone to forgetting in Row Permuted MNIST. Looking at the accuracy scale of Figure 3 (right), all the methods show severe degradation of performance compared to Permuted MNIST. This may be due to permuting of the correlated row blocks causing more weight changes in the network. After 10 tasks, UCL again achieved the highest average accuracy, 86.5%, in this experiment as well.

For a better understanding of our model, Figure 4 visualize the learned standard deviations of nodes in all layers as the training proceeds. After the model trained on task 1, we find that just a few of them become smaller than the initialized value of 0.06, and most of them become much larger in the first hidden layer. Interestingly, the uncertain nodes in layer 1 show a drastic decline of their standard deviations at a specific task as the learning progresses, which means the model had to make them certain for adapting to the new task. On the other hand, all the nodes in the output layer had to reduce their uncertainty as early as possible considering even a small randomness can lead to a totally different prediction. Most of the nodes in layer 2, in addition, do not show a monotonic tendency.

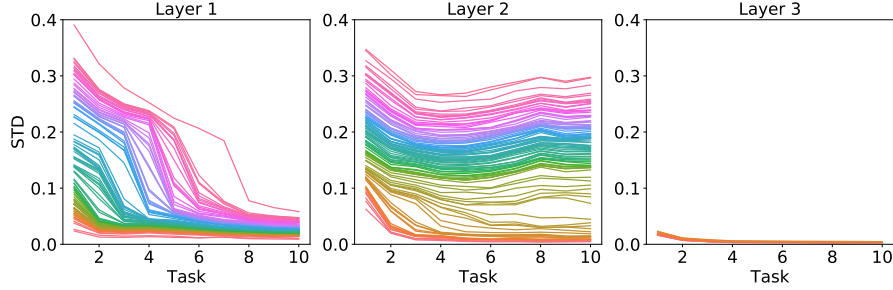

Figure 4: Standard deviation histogram in the Permuted MNIST experiment. We randomly selected 100 standard deviations for layer 1 and 2. In layer 3, all 10 nodes are shown.

This can be interpreted as many of them need not belong to a particular task. As a result, this gives the plasticity and gracefully forgetting trait of our UCL.

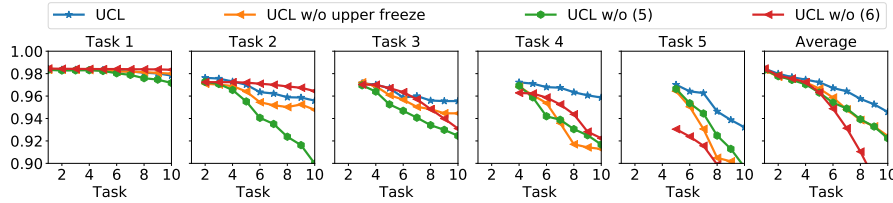

Figure 5: Ablation study in Permuted MNIST. Each line denotes the test accuracy.

We also carry out an ablation study on UCL's additional regularization terms. Figure 5 shows the results of three variations that lack one of the ingredients of the proposed UCL on Permuted MNIST. "UCL w/o upper freeze" stands for using $\mathbf{\Lambda}_{ij}^{(l)} = \sigma_{\text{init}}^{(l)}/\sigma_{t-1,i}^{(l)}$ in (4), and we observe regularizing the outgoing weights of an important node in UCL very important. "UCL w/o (5)" stands for the removing (5) from (7), and we clearly see the pruning heuristic based weight freezing is also very important. "UCL w/o (6)" stands for not using (6) and it shows that while the accuracy of Task 1 & 2 are even higher than UCL, but the accuracy drastically decreases after Task 3. This is because due to the rapid decrease of model capacity since "actively" learning weights reduce when (6) is not used.

**Split MNIST** We test also in the splitted dataset setting that each task consists of 2 consecutive classes of MNIST dataset. This benchmark was used in [30, 19] and has total 5 tasks. We used

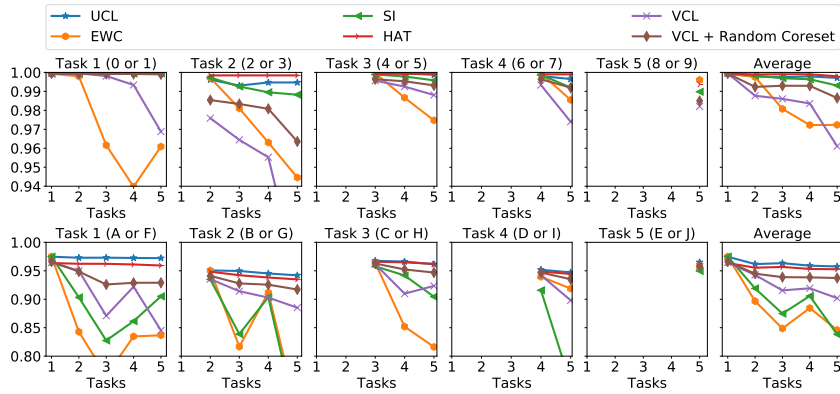

Figure 6: Experimental results on Split MNIST(top) and Split notMNIST(bottom)

multi-headed FNN hat has two hidden layers with 256 nodes and ReLU activations for all methods. In Figure 6 (top), we compare the test accuracy of each task together with the average accuracy over all observed tasks at the right end. UCL accomplishes the same 5 tasks average accuracy as HAT; 99.7%, which is slightly better than the results of SI and VCL with coreset, 99.0%, and 98.7%, respectively. Note UCL significantly outperforms EWC and VCL. We also point out that HAT makes a critical assumption to know the number of tasks *a priori*, while UCL need not.

**Split notMNIST** Here, we make an assessment on another splitted dataset tasks with notMNIST dataset, which has 10 character classes. We split the characters of notMNIST into 5 groups same as VCL[19]: A/F, B/G, C/H, D/I, and E/J. We used multi-headed FNN hat has four hidden layers with 150 nodes and ReLU activations for all methods. Unlike the previous experiments, SI shows similar results to EWC around 84% average accuracy, and VCL attains a better result of 90.1% (in Figure 6) (bottom). Our UCL again achieves a superior an outstanding result of 95.7%, that is higher than HAT and VCL with coreset: 95.2% and 93.7%, respectively.

**Split CIFAR and Omniglot** To check the effectiveness of UCL beyond the MNIST tasks, we experimented our UCL on three additional datasets, Split CIFAR-100, Split CIFAR10/100 and Omniglot. For Split CIFAR-100, each task consists of 10 consecutive classes of CIFAR-100, for Split CIFAR-10/100, we combined CIFAR-10 and Split CIFAR-100, and for Omniglot, each alphabet is treated as a single task, and we used all 50 alphabets. For Omniglot, as in [25], we rescaled all images to 28 × 28 and augmented the dataset by including 20 random permutations (rotations and shifting) for each image. For these datasets, unlike in the previous experiments using FNNs, we used deeper CNN architectures, in which the notion of *uncertainty* in the convolution layer is defined for each *channel* (i.e., filter). We used multi-headed outputs for all experiments, and 8 different random seed runs are averages for all datasets. The details of experiments using CNNs, including the architectures and hyperparameters, are given in the Supplementary Materials. In Figure 7, we compared UCL with EWC and SI and carried out extensive hyperparameter search for fair comparison. We did not compare with VCL since it did not have any results on vision datasets with CNN architectures.

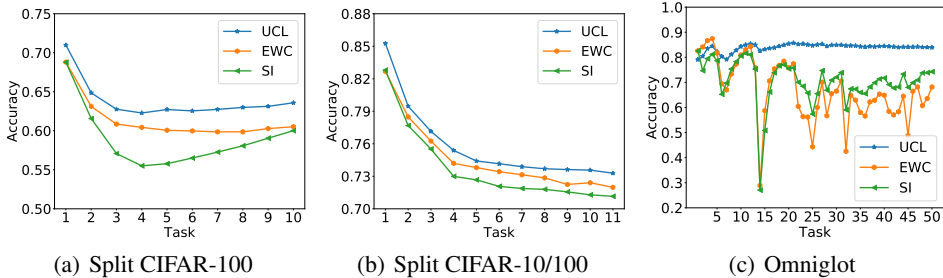

(a) Split CIFAR-100          (b) Split CIFAR-10/100          (c) Omniglot

Figure 7: Experiments on supervised learning using convolutional neural network

In Split CIFAR-100, EWC and SI achieve 60.5% and 60.0% respectively. However, UCL outperforms SI and EWC achieving 63.4%. In a slightly different task, Split CIFAR-10/100, which prevents overfitting on Split CIFAR-100 using a model pre-trained on CIFAR-10, UCL also outperforms baselines by achieving 73.2%. In Omniglot, although UCL becomes slightly unstable for the first task, it eventually achieves 83.9% average accuracy on all 50 tasks. However, EWC and SI only achieves 68.1% and 74.2% respectively, much lower than UCL. From above three results, we clearly observe UCL clearly outperforms the baselines for more diverse and sophisticated vision datasets and for deeper CNN architectures.

Table 1: The number of parameters used for each benchmark.

| Dataset\Methods | Vanilla | EWC | SI | HAT | VCL | UCL |
|---|---|---|---|---|---|---|
| Permuted MNIST | 478K | 1435K | 1435K | 486K | 1914K | 960K |
| Split MNIST | 270K | 808K | 808K | 272K | 1077K | 538K |
| Split notMNIST | 187K | 559K | 559K | 190K | 749K | 375K |
| Split CIFAR10/100 | 839K | 2467K | 2467K | - | - | 1655K |
| Omniglot | 1773K | 1995K | 1995K | - | - | 1884K |

**Comparison of model parameters** Table 1 shows the number of model parameters in each experiment. Vanilla stands for the base network architecture of all methods. It is shown that UCL has fewer parameters than other regularization-based approaches. Especially, UCL has almost half the number of VCL, which is based on the similar variational framework. Although HAT shows the least number of parameters, we stress it has the critical drawback of requiring to know the number of tasks *a priori*.

## 4.2 Reinforcement learning

Here, we also tested UCL for the continual reinforcement learning tasks. Roboschool [24] consists of 12 tasks and each task has a different shape of the state and continuous action space, and goal. From these tasks, we randomly chose eight tasks and sequentially learned each task (with 5 million update

steps) in the following order, {*Walker-HumanoidFlagrun-Hooper-Ant-InvertedDoublePendulum-Cheetah-Humanoid-InvertedPendulum*}. We trained a FNN model using PPO [24] as a training algorithm and selected EWC and Fine-tuning as baselines. All baselines were experimented in exactly the same condition, and we carried out an extensive hyperparameter search for fair comparison.

More experimental details, network architectures, and hyperparameters are given in the Supplementary Materials. Figure 8 shows the cumulative normalized rewards up to the learned task, and Figure 9 shows the normalized rewards for each task with vertical dotted lines showing the boundaries of the tasks. The normalization in the figures was done for each task with the maximum rewards obtained by EWC ($\lambda = 10$). The high cumulative sum thus corresponds to effectively combating the catastrophic forgetting (CF), and we note Fine-tuning mostly suffers from CF (e.g., Task2 or Task4). Note we show two versions of UCL, with different $\beta$ hyperparameter values. In Figure 8, we observe both versions of UCL significantly outperform both EWC and Fine-tuning. We believe the reason why EWC does not excel as in Figure 4B of the original EWC

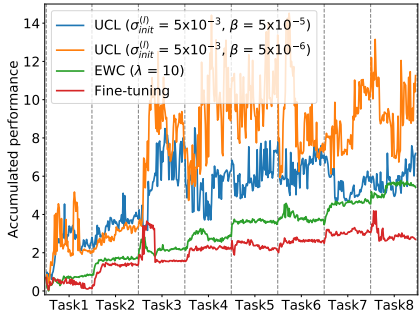

Figure 8: Cumulative normalized rewards.

paper [12] is because we consider pure continual learning setting, while [12] allows learning tasks multiple times in a recurring fashion. Moreover, a possible reason why UCL achieves such high rewards in RL setting may be due to the by-product of our weight sampling procedure; namely, the Gaussian perturbation of the weights for the variational inference enables an effective exploration of policies for RL as suggested in [21]. Figure 9 shows UCL overwhelmingly surpasses EWC particularly for Task1 and Task3 (by both achieving high rewards and not forgetting), and it contributes to the significant difference between EWC in Figure 8. We also experimentally checked the role of $\beta$ for *gracefully* forgetting; although UCL with $\beta = 5 \times 10^{-6}$ results in overall better rewards, with $\beta = 5 \times 10^{-5}$ does better in learning new tasks, *e.g.*, Task5/7/8, by adding more plasticity to the network. To the best of our knowledge, this result shows for the first time that the pure continual learning is possible for reinforcement learning with continuous action space and different observation shapes. We stress that there are very few algorithms in the literature that work well on both SL and RL continual learning setting, and our UCL is very competitive in that sense.

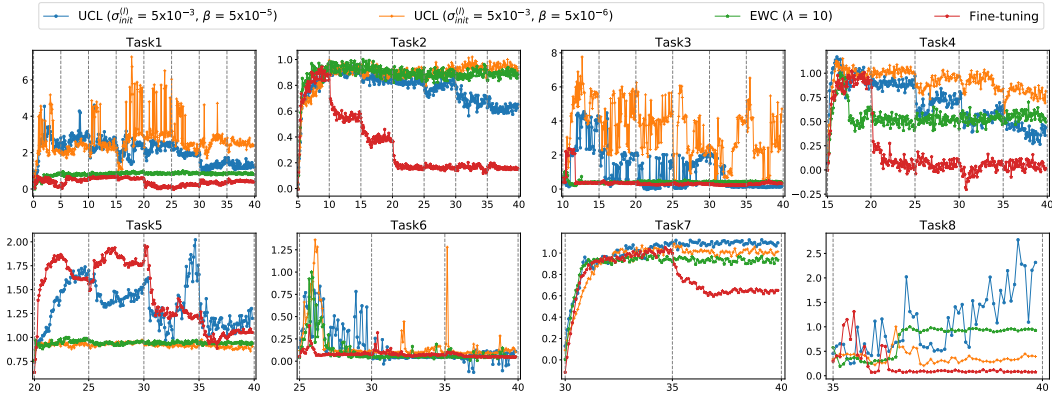

Figure 9: Normalized rewards for each task throughout learning of 8 RL tasks. Each task is learned with 5 million training steps. UCL excels in both not forgetting past tasks and learning new tasks.

## 5  Conclusion

We proposed UCL, a new uncertainty-based regularization method for overcoming catastrophic forgetting. We proposed the notion of node-wise uncertainty motivated from the Bayesian online learning framework and devised novel regularization terms for dealing with stability-plasticity dilemma. As a result, UCL convincingly outperformed other state-of-the-art baselines in both supervised and reinforcement learning benchmarks with much fewer additional parameters.

## Acknowledgements

This work is supported in part by ICT R&D Program [No. 2016-0-00563, Research on adaptive machine learning technology development for intelligent autonomous digital companion], AI Graduate School Support Program [No.2019-0-00421], and ITRC Support Program [IITP-2019-2018-0-01798] of MSIT / IITP of the Korean government.

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
