[Supplementary Material]

# Supplementary Materials for Uncertainty-regularized Continual Learning with Adaptive Regularization

**Hongjoon Ahn**[1]*, **Sungmin Cha**[2]*, **Donggyu Lee**[2] and **Taesup Moon**[1,2]
[1] Department of Artificial Intelligence, [2]Department of Electrical and Computer Engineering,
Sungkyunkwan University, Suwon, Korea 16419
{hong0805, csm9493, ldk308, tsmoon}@skku.edu

## 1 Derivation of Eq. (3)

Let's assume $q(\mathcal{W}|\boldsymbol{\theta}_t)$ and $q(\mathcal{W}|\boldsymbol{\theta}_{t-1})$ as below by the Gaussian mean-field approximation.

$$q(\mathcal{W}|\boldsymbol{\theta}_t) = N(\mathcal{W}|\boldsymbol{\mu}_t, \boldsymbol{\sigma}_t^2)$$

$$= \prod_{d=1}^{D} \frac{1}{\sqrt{2\pi\sigma_{t,d}^2}} \exp\left(-\frac{1}{2}\frac{(\mathcal{W}_d - \mu_{t,d})^2}{\sigma_{t,d}^2}\right)$$

$$q(\mathcal{W}|\boldsymbol{\theta}_t) = N(\mathcal{W}|\boldsymbol{\mu}_{t-1}, \boldsymbol{\sigma}_{t-1}^2)$$

$$= \prod_{d=1}^{D} \frac{1}{\sqrt{2\pi\sigma_{t-1,d}^2}} \exp\left(-\frac{1}{2}\frac{(\mathcal{W}_d - \mu_{t-1,d})^2}{\sigma_{t-1,d}^2}\right)$$

Then, $D_{KL}(q(\mathcal{W}|\boldsymbol{\theta}_t)||q(\mathcal{W}|\boldsymbol{\theta}_{t-1}))$ becomes as follows.

$$D_{KL}(q(\mathcal{W}|\boldsymbol{\theta}_t)||q(\mathcal{W}|\boldsymbol{\theta}_{t-1}))$$

$$= \int q(\mathcal{W}|\boldsymbol{\theta}_t) \log \frac{q(\mathcal{W}|\boldsymbol{\theta}_t)}{q(\mathcal{W}|\boldsymbol{\theta}_{t-1})} d\mathcal{W}$$

$$= \int N(\mathcal{W}|\boldsymbol{\mu}_t, \boldsymbol{\sigma}_t) \log \frac{N(\mathcal{W}|\boldsymbol{\mu}_t, \boldsymbol{\sigma}_t^2)}{N(\mathcal{W}|\boldsymbol{\mu}_{t-1}, \boldsymbol{\sigma}_{t-1}^2)} d\mathcal{W}$$

$$= \sum_{d=1}^{D} \int N(\mathcal{W}|\boldsymbol{\mu}_t, \boldsymbol{\sigma}_t^2) \log \frac{N(\mathcal{W}_d|\mu_t, \sigma_t^2)}{N(\mathcal{W}_d|\mu_{t-1,d}, \sigma_{t-1,d}^2)} d\mathcal{W}$$

$$= \sum_{d=1}^{D} \int N(\mathcal{W}_d|\mu_{t,d}, \sigma_{t,d}^2) \log \frac{N(\mathcal{W}_d|\mu_{t,d}, \sigma_{t,d}^2)}{N(\mathcal{W}_d|\mu_{t-1,d}, \sigma_{t-1,d}^2)} d\mathcal{W}_d \left(\int \prod_{i \neq d}^{D} N(\mathcal{W}_i|\mu_{t,i}, \sigma_{t,i}^2) d\mathcal{W}_i\right)$$

$$= \sum_{d=1}^{D} \int N(\mathcal{W}_d|\mu_{t,d}, \sigma_{t,d}) \log \frac{N(\mathcal{W}_d|\mu_{t,d}, \sigma_{t,d})}{N(\mathcal{W}_d|\mu_{t-1,d}, \sigma_{t-1,d})} d\mathcal{W}_d$$

$$= \sum_{d=1}^{D} \left( \int N(\mathcal{W}_d|\mu_{t,d}, \sigma_{t,d}^2) \log N(\mathcal{W}_d|\mu_{t,d}, \sigma_{t,d}^2) d\mathcal{W}_d \right. \tag{1}$$

$$\left. - \int N(\mathcal{W}_d|\mu_{t,d}, \sigma_{t,d}^2) \log N(\mathcal{W}_d|\mu_{t-1,d}, \sigma_{t,d}^2) d\mathcal{W}_d \right) \tag{2}$$

---

For simplicity, we decompose $D_{KL}(q(\mathcal{W}|\boldsymbol{\theta}_t)||q(\mathcal{W}|\boldsymbol{\theta}_{t-1}))$ as (1) and (2). First, the closed form of the integral in (1) becomes as follows.

$$\int N(\mathcal{W}|\mu_{t,d}, \sigma_{t,d}^2) \log N(\mathcal{W}|\mu_{t,d}, \sigma_{t,d}^2) d\mathcal{W}_d$$

$$= \int N(\mathcal{W}|\mu_{t,d}, \sigma_{t,d}^2) \log \frac{1}{\sqrt{2\pi\sigma_{t,d}^2}} \exp\left(-\frac{1}{2}\frac{(\mathcal{W}_d - \mu_{t,d})^2}{\sigma_{t,d}^2}\right) d\mathcal{W}_d$$

$$= \int N(\mathcal{W}_d|\mu_{t,d}, \sigma_{t,d}^2) \log \frac{1}{\sqrt{2\pi\sigma_{t,d}^2}} \exp\left(-\frac{1}{2}\frac{(\mathcal{W}_d - \mu_{t,d})^2}{\sigma_{t,d}^2}\right) d\mathcal{W}_d$$

$$= \log \frac{1}{\sqrt{2\pi\sigma_{t,d}^2}} + \int N(\mathcal{W}_d|\mu_{t,d}, \sigma_{t,d}^2)\left(-\frac{1}{2}\frac{(\mathcal{W}_d - \mu_{t,d})^2}{\sigma_{t,d}^2}\right) d\mathcal{W}_d$$

$$= \log \frac{1}{\sqrt{2\pi\sigma_{t,d}^2}} - \frac{1}{2\sigma_{t,d}^2} Var[\mathcal{W}_d]_{N(\mu_{t,d},\sigma_{t,d}^2)}$$

$$= -\frac{1}{2}\log 2\pi\sigma_{t,d}^2 - \frac{1}{2} \tag{3}$$

Next, the integral in (2) becomes as follows.

$$\int N(\mathcal{W}|\mu_{t,d}, \sigma_{t,d}^2) \log N(\mathcal{W}|\mu_{t-1,d}, \sigma_{t-1,d}^2) d\mathcal{W}_d$$

$$= \int N(\mathcal{W}|\mu_{t,d}, \sigma_{t,d}^2) \log \frac{1}{\sqrt{2\pi\sigma_{t-1,d}^2}} \exp\left(-\frac{1}{2}\frac{(\mathcal{W}_d - \mu_{t-1,d})^2}{\sigma_{t-1,d}^2}\right) d\mathcal{W}_d$$

$$= \int N(\mathcal{W}_d|\mu_{t,d}, \sigma_{t,d}^2) \log \frac{1}{\sqrt{2\pi\sigma_{t-1,d}^2}} \exp\left(-\frac{1}{2}\frac{(\mathcal{W}_d - \mu_{t-1,d})^2}{\sigma_{t-1,d}^2}\right) d\mathcal{W}_d$$

$$= \log \frac{1}{\sqrt{2\pi\sigma_{t-1,d}^2}} + \int N(\mathcal{W}_d|\mu_{t,d}, \sigma_{t,d}^2)\left(-\frac{1}{2}\frac{(\mathcal{W}_d - \mu_{t-1,d})^2}{\sigma_{t-1,d}^2}\right) d\mathcal{W}_d$$

$$= \log \frac{1}{\sqrt{2\pi\sigma_{t-1,d}^2}} - \frac{1}{2\sigma_{t-1,d}^2} E[(\mathcal{W}_d - \mu_{t-1,d})^2]_{N(\mathcal{W}_d|\mu_{t,d},\sigma_{t,d}^2)}$$

$$= -\frac{1}{2}\log 2\pi\sigma_{t-1,d}^2 - \frac{1}{2\sigma_{t-1,d}^2}\left(\sigma_{t,d}^2 + (\mu_{t,d} - \mu_{t-1,d})^2\right) \quad (\because E[(X-a)^2] = Var[X] + (E[X]-a)^2)$$

$$\tag{4}$$

Therefore, combining (3) and (4), $D_{KL}(q(\mathcal{W}|\boldsymbol{\theta}_t)||q(\mathcal{W}|\boldsymbol{\theta}_{t-1}))$ becomes

$$D_{KL}(q(\mathcal{W}|\boldsymbol{\theta}_t)||q(\mathcal{W}|\boldsymbol{\theta}_{t-1})) = \sum_{d=1}^{D}\left[\frac{1}{2\sigma_{t-1,d}^2}(\mu_{t,d} - \mu_{t-1,d})^2 + \frac{1}{2}\left(\frac{\sigma_{t,d}^2}{\sigma_{t-1,d}^2} - \log\frac{\sigma_{t,d}^2}{\sigma_{t-1,d}^2}\right)\right] + C$$

$$= \frac{1}{2}\left\|\frac{\boldsymbol{\mu}_t - \boldsymbol{\mu}_{t-1}}{\boldsymbol{\sigma}_t}\right\|_2^2 + \frac{1}{2}\mathbf{1}^\top\left\{\left(\frac{\boldsymbol{\sigma}_t^2}{\boldsymbol{\sigma}_{t-1}^2} - \log\frac{\boldsymbol{\sigma}_t^2}{\boldsymbol{\sigma}_{t-1}^2}\right)\right\} + C \tag{5}$$

where $\mathbf{1}$ stands for all-1 vector with $D$ dimensions. Therefore, (5) becomes the same as Eq. (3) in main paper when we decompose into the terms for each layer in the network.

## 2    Detailed explanation on initializing standard deviation

As mentioned in Section 4.1, due to using much deeper architecture in convolutional neural networks, we used another initialization technique which is adaptive to model architecture. Our initialization method is highly motivated by [1], and the notations on derivations are similar to [1]. As discussed in [1], we mainly consider ReLU activation for deriving the initialization method.

**Forward propagation case** Let assume a forward propagation in $l$-th layer is

$$\mathbf{y}_l = \boldsymbol{\mathcal{W}}_l \mathbf{h}_{l-1} + \mathbf{b}_l,$$

in which, $\boldsymbol{\mathcal{W}}_l$ is sampled from some prior distribution $p(\boldsymbol{\mathcal{W}}|\alpha)$ which is a symmetric distribution with zero mean, $\mathbf{h}_{l-1}$ is the activation value of the previous layer, and $\mathbf{b}_l$ is bias. As in [1], we assume that all activation values in $\mathbf{h}_{l-1}$ are i.i.d., and $\mathbf{h}_{l-1}$ and $\boldsymbol{\mathcal{W}}_l$ are independent. Since $\boldsymbol{\mathcal{W}}_l$ has zero mean,

$$\begin{aligned}
\mathbf{Var}[y_l] &= n_l \cdot \mathbf{Var}[w_l \cdot h_{l-1}] \\
&= n_l \cdot \mathbf{Var}[w_l] \cdot \mathbf{E}[h_{l-1}^2],
\end{aligned} \tag{6}$$

where $n_l$ is the number of nodes in the previous layer. Note that in convolutional neural network, $n_l$ is $k^2 c$-by-1 vector, in which c is the number of channels, and $k$ is the filter size of the layer. If we assume $\mathbf{b}_l = 0$, then $y_{l-1}$ has zero mean and symmetric distribution. Therefore, considering ReLU activation, $\mathbf{E}[h_{l-1}^2]$ can be replaced by $\frac{1}{2}\mathbf{Var}[y_{l-1}]$ in (6). Putting all together with $L$ layers, we have

$$\mathbf{Var}[y_L] = \mathbf{Var}[y_1] \cdot \left( \prod_{l=2}^{L} \frac{1}{2} n_l \cdot \mathbf{Var}[w_l] \right). \tag{7}$$

Note that, in (7), the softmax output of the network can be increased or decreased exponentially, which leads to unstable training. To avoid this problem, $\mathbf{Var}[y_L]$ should be a constant. Hence, a sufficient condition for setting $\mathbf{Var}[y_L]$ constant is

$$\begin{aligned}
\mathbf{Var}[w_l] &= \mathbf{Var}[\mu_l + \sigma_{\text{init}}^{(l)}\epsilon] \\
&= \mathbf{Var}[\mu_l] + (\sigma_{\text{init}}^{(l)})^2 \\
&= \frac{2}{n_l}.
\end{aligned}$$

To maintain this condition, we initialize $\mathbf{b}_l = 0$, $(\sigma_{\text{init}}^{(l)})^2 = r \cdot \frac{2}{n_l}$, and $\mathbf{Var}[\mu_l] = (1-r) \cdot \frac{2}{n_l}$, in which $0 \leq r \leq 1$ is a hyperparameter.

**Backward propagation case** For backpropagation, the gradient of loss in convolution layer is

$$\Delta \mathbf{h}_{l-1} = \hat{\boldsymbol{\mathcal{W}}}_l \Delta \mathbf{y}_l, \tag{8}$$

where $\Delta \mathbf{h}_{l-1}$ is $\frac{\partial L}{\partial \mathbf{h}_{l-1}}$, $\Delta \mathbf{y}_l$ is $\frac{\partial L}{\partial \mathbf{h}_{l-1}}$, and $\hat{\boldsymbol{\mathcal{W}}}_l$ is transposed version of $\boldsymbol{\mathcal{W}}_l$. Similar as forward propagation case, we can compute the variance of the gradient in (8), which is

$$\begin{aligned}
\mathbf{Var}[\Delta h_{l-1}] &= \hat{n}_l \mathbf{Var}[w_l] \mathbf{Var}[\Delta y_l] \\
&= \frac{1}{2} \hat{n}_l \mathbf{Var}[w_l] \mathbf{Var}[\Delta h_l].
\end{aligned}$$

Note that we denote the number of nodes in the upper layer as $\hat{n}_l$. In convolution layer, $\hat{n}_l$ is $k^2 d$-by-1 vector. As mentioned in (7), the variance of gradient in the input layer is

$$\mathbf{Var}[\Delta h_1] = \mathbf{Var}[\Delta h_L] \left( \prod_{l=1}^{L} \mathbf{Var}[w_l] \right). \tag{9}$$

To avoid the signals exponentially increasing or decreasing, we set the variance of the weights as

$$\begin{aligned}
\mathbf{Var}[w_l] &= \mathbf{Var}[\mu_l + \sigma_{\text{init}}^{(l)}\epsilon] \\
&= \mathbf{Var}[\mu_l] + (\sigma_{\text{init}}^{(l)})^2 \\
&= \frac{2}{\hat{n}_l}.
\end{aligned}$$

Same as in forward propagation case, we initialize $\mathbf{b}_l = 0$, $(\sigma_{\text{init}}^{(l)})^2 = r \cdot \frac{2}{\hat{n}_l}$ and $\mathbf{Var}[\mu_l] = (1-r) \cdot \frac{2}{\hat{n}_l}$, in which the range of $r$ is $0 \leq r \leq 1$.

When carrying out experiments, we empirically find that initializing $(\sigma_{\text{init}}^{(l)})^2 = r \cdot \frac{2}{\hat{n}_l}$ for convolution layers and $(\sigma_{\text{init}}^{(l)})^2 = r \cdot \frac{2}{n_l}$ for fully connected layers achieves the best performance.

## 3 Detailed explanation on Eq. (6)

The modified term of term (b) by adding (6) is

$$\sum_{l=1}^{L} \frac{1}{2}\mathbf{1}^{\top}\left(\left(\frac{\boldsymbol{\sigma}_t^{(l)}}{\boldsymbol{\sigma}_{t-1}^{(l)}}\right)^2 - \log\left(\frac{\boldsymbol{\sigma}_t^{(l)}}{\boldsymbol{\sigma}_{t-1}^{(l)}}\right)^2 + (\boldsymbol{\sigma}_t^{(l)})^2 - \log(\boldsymbol{\sigma}_t^{(l)})^2\right). \qquad (10)$$

To verify the optimal point of (10), we first convert (10) to generalized form, which is

$$\sum_{l=1}^{L} \frac{1}{2}\mathbf{1}^{\top}\left(\left(\frac{\boldsymbol{\sigma}_t^{(l)}}{\boldsymbol{\sigma}_{t-1}^{(l)}}\right)^2 - \log\left(\frac{\boldsymbol{\sigma}_t^{(l)}}{\boldsymbol{\sigma}_{t-1}^{(l)}}\right)^2 + (\frac{\boldsymbol{\sigma}_t^{(l)}}{\{\boldsymbol{p}^{(l)}\}^2}) - \log(\frac{\boldsymbol{\sigma}_t^{(l)}}{\{\boldsymbol{p}^{(l)}\}^2})\right), \qquad (11)$$

where $\boldsymbol{p}^{(l)}$ is any vector satisfying $p_i^{(l)} \gg \sigma_{t-1,i}^{(l)}$ for all $i$. Since (11) is a convex function, the gradient with respect to $\boldsymbol{\sigma}_t^{(l)}$ at optimal point is $\mathbf{0}$. The gradient of (11) is

$$\nabla_{\boldsymbol{\sigma}_t^{(l)}}\left(\sum_{l=1}^{L} \frac{1}{2}\mathbf{1}^{\top}\left(\left(\frac{\boldsymbol{\sigma}_t^{(l)}}{\boldsymbol{\sigma}_{t-1}^{(l)}}\right)^2 - \log\left(\frac{\boldsymbol{\sigma}_t^{(l)}}{\boldsymbol{\sigma}_{t-1}^{(l)}}\right)^2 + (\frac{\boldsymbol{\sigma}_t^{(l)}}{\{\boldsymbol{p}^{(l)}\}^2}) - \log(\frac{\boldsymbol{\sigma}_t^{(l)}}{\{\boldsymbol{p}^{(l)}\}^2})\right)\right)$$

$$= \frac{\boldsymbol{\sigma}_t^{(l)}}{\{\boldsymbol{\sigma}_{t-1}^{(l)}\}^2} - \frac{1}{\boldsymbol{\sigma}_t^{(l)}} + \frac{\boldsymbol{\sigma}_t^{(l)}}{\{\boldsymbol{p}^{(l)}\}^2} - \frac{1}{\boldsymbol{\sigma}_t^{(l)}}$$

$$= \boldsymbol{\sigma}_t^{(l)} \odot \left(\frac{1}{\{\boldsymbol{\sigma}_{t-1}^{(l)}\}^2} + \frac{1}{\{\boldsymbol{p}^{(l)}\}^2} - \frac{2}{\{\boldsymbol{\sigma}_t^{(l)}\}^2}\right). \qquad (12)$$

The point which makes (12) equal to $\mathbf{0}$ is the optimal point. Therefore, the optimal point is

$$\boldsymbol{\sigma}_t^{(l)} = \sqrt{\frac{2}{\frac{1}{\{\boldsymbol{\sigma}_{t-1}^{(l)}\}^2} + \frac{1}{\{\boldsymbol{p}^{(l)}\}^2}}} = \sqrt{\frac{2 \cdot \{\boldsymbol{p}^{(l)}\}^2 \odot \{\boldsymbol{\sigma}_{t-1}^{(l)}\}^2}{\{\boldsymbol{\sigma}_{t-1}^{(l)}\}^2 + \{\boldsymbol{p}^{(l)}\}^2}} \approx \sqrt{2}\boldsymbol{\sigma}_{t-1}^{(l)}(\because \{\boldsymbol{p}^{(l)}\}^2 \gg \{\boldsymbol{\sigma}_{t-1}^{(l)}\}^2), \qquad (13)$$

in which $\gg$ is element-wise comparison. In (13), selecting any vector $\boldsymbol{p}^{(l)}$ which satisfies $\boldsymbol{p}^{(l)} \gg \boldsymbol{\sigma}_{t-1}^{(l)}$ can achieve $\boldsymbol{\sigma}_t^{(l)} \approx \sqrt{2}\boldsymbol{\sigma}_{t-1}^{(l)}$. Therefore, for simplicity, we select $\boldsymbol{p}^{(l)} = 1$ for all layers.

## 4 Additional experimental results

We carry out additional experiments to see the effect of adaptive initialization in the fully connected layer network. In Figure 1 and 2, "UCL constant" represents initializing $\sigma_{\text{init}}^{(l)} = c$ and "UCL adaptive" represents initializing $\sigma_{\text{init}}^{(l)}$ as adaptive to layer size.

Figure 1: Additional experimental results on various Permuted MNIST with single head.

We also carry out an additional ablation study in Split CIFAR10/100 using convolutional neural network. Figure 3 shows the additional results on the ablation study in Split CIFAR 10/100. In Figure 3, we initialized $\{\sigma_{\text{init}}^{(l)}\}_{l=1}^{L}$ adaptive to layer size. The legend on Figure 3 is same as in the manuscript.

Figure 2: Additional experimental results on Split MNIST(top) and Split notMNIST(bottom)

In Figure 3, we can observe that "UCL w/o upper freeze" does not effectively prevent catastrophic forgetting in early tasks. However, since it gives low regularization strength on outgoing weights, a large number of "active learners" tend to help train future tasks effectively, which leads to achieving high average accuracy. "UCL w/o (5)" has much lower retention capability on Task 1 than original UCL. However, in terms of average accuracy, both of them achieve almost the same accuracy. "UCL w/o (6)" shows that the accuracy of Task 1 is even higher than UCL. However, since it does not have any gracefully forgetting techniques, the accuracy drastically decreases after Task 1.

Figure 3: Ablation study in Split CIFAR10/100 using adaptive initialization.

# 5 Implementation details

## 5.1 Supervised learning

### 5.1.1 Training details

**Permuted MNIST / Row Permuted MNIST**

We trained all of our baselines with mini batch size of 256 for 100 epochs other than VCL, which used 200 epochs. We also optimized them with learning rate 0.001 by Adam optimizer[2]. But for HAT, we updated it by stochastic gradient descent(SGD) with learning rate 0.05. For EWC, the Fisher information matrix were computed using all training samples of a task. Regularization hyperparameters are compared as below :

- UCL
  - $\beta$ : {0.0001, 0.001, 0.01, **0.02 (best for Row Permuted MNIST)**, **0.03 (best for Permuted MNIST)**}
- EWC

- $\lambda$ : {40, **400 (best)**, 4000, 40000}
- SI
  - $c$ : {0.01, **0.03 (best)**, 0.1, 0.3, 0.5, 0.7, 1.0}
- HAT
  - $c$ : {0.25, 0.5, **0.75( best)**, 1.0}
- VCL - not needed

## Split MNIST

We use the whole training dataset of a task for a batch size of VCL, and trained it for 120 epochs. The others are trained in the same way as previous experiment. Regularization hyperparameters are compared as below :

- UCL
  - $\beta$ : {**0.0001 (best)**, 0.001, 0.01, 0.02, 0.03 (best)}
- EWC
  - $\lambda$ : {40, 400, **4000 (best)**, 40000}
- SI
  - $c$ : {0.01, 0.03, 0.1, 0.3, 0.5, 0.7, **1.0 (best)**}
- HAT
  - $c$ : {0.25, 0.5, **0.75 (best)**, 1.0}
- VCL - not needed

## Split notMNIST

The training settings are equal to those in Split MNIST. Hyperparameters are compared as below :

- UCL
  - $\beta$ : {0.0001, **0.001 (best)**, 0.01, 0.02, 0.03(best)}
- EWC
  - $\lambda$ : {40, 400, **4000 (best)**, 40000}
- SI
  - $c$ : {0.01, 0.03, 0.1, **0.3 (best)**, 0.5, 0.7, 1.0}
- HAT
  - $c$ : {0.25, 0.5, **0.75 (best)**, 1.0}
- VCL - not needed

## Split CIFAR-10/100 / Split CIFAR-100

Same as in Permuted MNIST experiment, we trained UCL and all our baselines with mini-batch size of 256 for 100 epochs. We also optimized them with learning rate 0.001 using the Adam optimizer [2]. The network architecture used for Split CIFAR-10/100 and Split CIFAR-100 experiments is given in Table 1. The hyperparameters used in Split CIFAR-10/100 & Split CIFAR-100 experiments are compared as below :

- UCL
  - $\beta$ : {0.0001, **0.0002 (best for CIFAR-10/100)**, 0.001, **0.0002 (best for CIFAR-100)**}
  - $r$ : {0.5, **0.125 (best)**}
  - $lr(\sigma)$ : {**0.01 (best)**, 0.02}
- EWC
  - $\lambda$ : {25, 50, 75, 100, 250, 500, 750, 1000, 2500, 5000, 7500, **10000 (best for CIFAR-100)**, **25000 (best for CIFAR-10/100)**, 50000, 75000, 100000}
- SI
  - $c$ : {0.25, 0.3, 0.35, 0.4, 0.45, 0.5, 0.55, 0.6, 0.65, **0.7 (best for CIFAR-10/100)**, 0.75, 0.8, 0.85, 0.9, 0.95, **1.0 (best for CIFAR-100)**}

Table 1: Network architecture for Split CIFAR-10/100 and Split CIFAR-100

| Layer | Channel | Kernel | Stride | Padding | Dropout |
|---|---|---|---|---|---|
| 32×32 input | 3 | | | | |
| Conv 1 | 32 | 3×3 | 1 | 1 | |
| Conv 2 | 32 | 3×3 | 1 | 1 | |
| MaxPool | | | 2 | 0 | 0.25 |
| Conv 3 | 64 | 3×3 | 1 | 1 | |
| Conv 4 | 64 | 3×3 | 1 | 1 | |
| MaxPool | | | 2 | 0 | 0.25 |
| Conv 5 | 128 | 3×3 | 1 | 1 | |
| Conv 6 | 128 | 3×3 | 1 | 1 | |
| MaxPool | | | 2 | 1 | 0.25 |
| Dense 1 | 256 | | | | |
| Task 1 : Dense 10 | | | | | |
| $\cdots$ | | | | | |
| Task $i$ : Dense 10 | | | | | |

## Omniglot

Same as in Split CIFAR-10/100 experiment, we trained UCL and all our baselines with mini batch size of 256 for 100 epochs. We also optimized them with learning rate 0.001 using the Adam optimizer[2]. The network architecture used in Omniglot experiments is given in Table 2. Since the number of classes for each task is different, we denoted the classes of $i$th task as $C_i$. The hyperparameters used

Table 2: Network architecture on Omniglot

| Layer | Channel | Kernel | Stride | Padding | Dropout |
|---|---|---|---|---|---|
| 28×28 input | 1 | | | | |
| Conv 1 | 64 | 3×3 | 1 | 0 | |
| Conv 2 | 64 | 3×3 | 1 | 0 | |
| MaxPool | | | 2 | 0 | 0 |
| Conv 3 | 64 | 3×3 | 1 | 0 | |
| Conv 4 | 64 | 3×3 | 1 | 0 | |
| MaxPool | | | 2 | 0 | 0 |
| Task 1 : Dense $C_1$ | | | | | |
| $\cdots$ | | | | | |
| Task $i$ : Dense $C_i$ | | | | | |

in Omniglot experiments are compared as below :

- UCL
  - $\beta$ : {**0.00001 (best)**, 0.0001, 0.001, 0.01}
  - $r$ : {**0.5 (best)**, 0.125}
  - $lr(\sigma)$ : {0.01, **0.02 (best)**}
- EWC
  - $\lambda$ : {25, 50, 75, 100, 250, 500, 750, 1000, 2500, 5000, 7500, 10000, 25000, 50000, 75000, **100000 (best)**}
- SI
  - $c$ : {0.25, 0.3, 0.35, 0.4, 0.45, 0.5, 0.55, 0.6, 0.65, 0.7, 0.75, 0.8, 0.85, 0.9, 0.95, **1.0 (best)**}

### 5.1.2 UCL for convolutional neural networks

Figure 4 shows the implementation of UCL for convolutional neural networks (CNN). Instead of giving uncertainty on nodes, we define uncertainty for convolution channels. Colored channels and filters denote important channels and highly regularized filters due to Eq.(4) in the paper. Suppose when training Task 1 is finished, assume the orange colored channels represent channels identified to be important (*i.e.*, certain) for Task 1. Based on Eq.(4) in the manuscript, the whole filters which mainly contribute to making orange colored channels and filter coefficients which use the orange colored channels as inputs getting high regularization strengths for Task 1. Then, after training Task

Figure 4: Implementation details on convolutional neural network.

2, assume the green-colored channels represent channels specific to Task 2. Similar to the case of Task 1, filters connected with green-colored channels are dedicated to Task 2. However, since filters dedicated to Task 1 already get high regularization strengths, these filters tend to keep their values. Therefore, except for filters which already get high regularization strength for Task 1, filters connected to the green-colored channels are expected to be dedicated to Task 2. And gray shaded filters that do not belong to neither Task 1 nor Task 2 are active learners which can be trained future tasks actively.

## 5.2 Reinforcement learning

### 5.2.1 Information on the selected eight tasks and our FNN model

Table 3: Details on environments.

| Task number | Task name | Observation shape | Action shape | Goal |
|---|---|---|---|---|
| 1 | Walker2d | (22,) | (3,) | Make robot run as fast as possible |
| 2 | HumanoidFlagrun | (44,) | (17,) | Make a 3D humanoid robot run towards a target |
| 3 | Hopper | (15) | (3,) | Make the hopper hop as fast as possible |
| 4 | Ant | (28) | (8,) | Make the creature walk as fast as possible |
| 5 | InvertedDoublePendulum | (3,) | (1,) | Keep a pole upright by moving the 1-D cart |
| 6 | HalfCheetah | (28,) | (8,) | Make the creature walk as fast as possible |
| 7 | Humanoid | (22,) | (6,) | Make robot run as fast as possible |
| 8 | InvertedPendulum | (3,) | (1,) | Keep a pole upright by moving the 1-D cart |

Table 3 shows details on environments that we used in the experimental section on reinforcement learning. We trained UCL and each baseline using eight tasks in Table 3. We used two fully connected hidden layers with 16 nodes, one input layer and multiple output layers. To initialize the model parameters of baselines, we used *He initialization*. The number of nodes in the input layer is 44, which is the maximum size of the state space in selected eight tasks. Depending on the tasks, unused areas are filled with zero-masks. Each output layer is equal to the size of the action space of each task, and we used a multivariate Gaussian distribution layer, which learns mean and standard deviation for a continuous action, as an output layer. We evaluated each task for 10 episodes and report the averaged rewards.

### 5.2.2 Hyperparameters of PPO

We used PPO (Proximal Policy Optimization) to train a model for reinforcement learning. Figure 4 shows hyperparameters we used and these hyperparameters are applied to the all baselines in reinforcement learning experiments equally.

Table 4: Details on hyperparameters of PPO.

| Hyperparameters | Value |
|---|---|
| # of steps of each task | 5m |
| # of processes | 128 |
| # of steps per iteration | 64 |
| PPO epochs | 10 |
| entropy coefficient | 0 |
| value loss coefficient | 0.5 |
| $\gamma$ for accumulated rewards | 0.99 |
| $\lambda$ for GAE | 0.95 |
| mini-batch size | 64 |

## 5.3 Additional experimental results with $\sigma_{\text{init}}^{(l)} = 1 \times 10^{-3}$

Figure 5: Cumulative normalized rewards.

The combination of $\sigma_{\text{init}}^{(l)}$ and $\beta$ controls the model capacity for continual learning, and also $\sigma_{\text{init}}^{(l)}$ is more involved in the initial capacity of model in view of the level of uncertainty. As a result, when we set appropriate $\sigma_{\text{init}}^{(l)}$, as in the paper, we get higher accumulated rewards. As in an additional experiments, Figure 5 and 6 shows the experimental results with $\sigma_{\text{init}}^{(l)} = 1 \text{x} 10^{-3}$ for UCL. The results of other baselines are the same as the paper and all experimental results of each task are normalized by the maximum and minimum value of EWC. Compared to Figure 9 in the manuscript, Figure 6 shows similar results: UCL model with small $\beta$ can overcome catastrophic forgetting effectively. However, we also find two different results from this experiment. First, compared to Figure 9 in the manuscript, Task 1 and 3 get much lower rewards. We believe that giving smaller initial uncertainty on the nodes limits the exploration capacity of the model, hence, results in the lower rewards. Another different result is in Task 8. From Figure 9 in the manuscript, the result of different $\beta$ shows that we can control the level of gracefully forgetting by selecting $\beta$. However, Task 8 in Figure 6 has a difficulty to learn a new task in both $\beta$ cases. This result shows that, if the uncertainty of the model is small, the model will have difficulty to learn a new task even if the model forgets gracefully. In conclusion, we stress that it is important to set $\sigma_{\text{init}}^{(l)}$ and $\beta$ values properly to achieve successful continual reinforcement learning in both goals, overcoming catastrophic forgetting by selecting $\beta$ and achieving highly enough rewards by selecting $\sigma_{\text{init}}^{(l)}$.

Figure 6: Normalized rewards for each task throughout learning of 8 RL tasks.