[Reviews · NeurIPS 2019]

Reviewer 1



Summary: The paper presents a regularization-based continual learning method, UCL, where during the training of the current task the parameters of the network are regularized based on their uncertainty in the previous tasks (less uncertainty means that a parameter is important and should not be altered in future tasks). Instead of measuring the uncertainty at the parameter level, as done in the earlier works (e.g.) Variational Continual Learning (VCL), the authors propose to measure uncertainty over the neurons resulting in less number of learnable parameters (mean and variances) to store. To compute the neurons uncertainty, UCL imposes a constraint that all the weights going into a neuron share the same/ common variance. To learn the parameters, a variational objective is used where the authors cleverly opened up the KL term in the ELBO and play with it to impose constraints on the variances of different neurons. The results are reported on MNIST benchmarks and RL tasks. Positives: 1 - The paper is well-written and the final model is well-motivated. 2 - The KL term (Eq. 3) gives interesting insights and the authors exploit the insights carefully to impose different restrictions on the parameters update to avoid forgetting while keeping the network capacity sufficient to learn new tasks. 3 - I quite like the ablation performed on different facets of the objective function in Figure 5. Negatives: I overall quite liked the paper. While the method is well-motivated, I am concerned with the experiments (especially the supervised learning experiment). The reporting of results on only MNIST is a growing concern in the CL community and there’s enough literature available showing that the MNIST is not a very good benchmark to test whether a continual learning method is working or not. While most of the other benchmarks that people deploy (e.g.) Split-[CIFAR, CUB, miniImageNet] have their own problems, they at least make the problem setup more complex and interesting. At the very least, I expect the authors to try their method on these benchmarks and see how their method fare compared to others. I am leaning towards borderline pending the requested experiments. Post-rebuttal: I read the authors' response and they adequately answer my main concern of testing their approach only on the MNIST for supervised learning experiments. As for the novelty, I still believe the proposed interpretation of the KL term in the ELBO is original and gives interesting insights to the online bayesian learning frameworks. I will recommend acceptance.

Reviewer 2



Post-rebuttal: I have read the rebuttal. I think the rebuttal has sufficiently addressed my questions. With the new experiments for supervised learning and reinforcement learning, I think the paper is much stronger now. So, I will vote for accepting this paper. --------- - Originality: I think the proposed method is novel. - Quality: 1. I find the proposed regularization term a bit messy with 3 components added to the original objective function. However, the paper explains the reasons behind the modifications well. 2. On L206: did you sample only one sample weight once at the beginning of the optimization process? Or did you sample one sample weight at every iteration of the optimization? From Eq 7, it seems that you took the first approach, which I find very strange that it could work at all since the log-likelihood term would not depend on the parameters of the model in this case. - Clarity: The paper is mostly clear, except that the authors used too many "so-called" in Sec 3.1, which gives the impression that they don't agree with the names of the methods in literature. Please consider fixing them if it is not intentional. - Significance: I think the contributions in this paper is reasonably significant.

Reviewer 3



This paper changes the way of the regularization of the VCL algorithm based on some heuristic intuitions. The idea of maintaining the uncertainty for each hidden node is actually interesting. And it also makes more sense to give high regularization strength when either of the nodes it connects has low uncertainty. However, I think the proposed method in this paper focuses on the modification on the regularization which only provides incremental improvement with respect to VCL. The ideology behind the proposed method is not well explained from a theoretical viewpoint. The effectiveness of the proposed algorithm cannot be verified just through a two-hidden-layers fully connected network. Since this paper uses two hyperparameters to control the uncertainty, it is important to include experimental results show the influence of them.

[Author Response · NeurIPS 2019]



|   (a) Split CIFAR-100   |   (b) Omniglot   |   (c) Accum. rewards for 8 RL tasks   |

Figure 1: Additional experiments on supervised learning and reinforcement learning tasks

**Supervised learning [R1,R3]** To check the effectiveness of UCL beyond the MNIST tasks, we experimented our UCL on two additional datasets, Split CIFAR-100 and Omniglot. For Split CIFAR-100, each task consists of 10 consecutive classes of CIFAR-100, and for Omniglot, each alphabet is treated as a single task, and we randomly sampled 10 tasks from all 50 alphabets. For Omniglot, we rescaled all images to $28 \times 28$ and augmented the dataset by including 20 random permutations (rotations and shifting) for each image. For both datasets, unlike the experiments in the manuscript, we used deeper CNN architectures, for which the notion of *uncertainty* in the convolution layer is defined for each *channel* (i.e., filter). For Split CIFAR-100, we used 6 $3 \times 3$ convolution layers with 32-32-64-64-128-128 channels and 2 dense layers with 2048 and 256 nodes, and for Omniglot, we used 4 $3 \times 3$ convolutional layers with 64 channels and 1 dense layer with 1024 nodes. We used multi-head outputs for both experiments, and 5 and 3 different random seed runs are averaged for Split CIFAR-100 and Omniglot, respectively. In Figure 1(a) and 1(b), we compared with EWC and SI and carried out extensive hyperparameter search for fair comparison. We did not compare with VCL since it did not have any results on vision datasets with CNN architecture. From the figures, we clearly observe that UCL outperforms the baselines for both tasks as well, stressing the effectiveness of UCL on diverse datasets. We could not carry out experiments on CUB and miniImageNet due to time constraint, and we will defer to the future work.

**Reinforcement learning [R1-R3]** We believe one of the important contributions of UCL is its strong performance in reinforcement learning setting. To that end, we conducted additional experiments on the Roboschool platform that expands the results in Section 4.2 of the manuscript. Namely, we randomly selected 8 tasks, {*Walker-HumanoidFlagrun-Hooper-Ant-InvertedDoublePendulum-Cheetah-Humanoid-InvertedPendulum*} and carried out continual learning (i.e., the past task data is not available once a new task is learned). We used two fully connected layers with 16 nodes and other hyperparameters were equal to the described in the manuscript. The hyperparamters were set to $\beta = 0.001$ and $\sigma_{\text{init}} = \{0.001, 0.005\}$ to show the influence of $\sigma_{\text{init}}$. Figure 1(c) shows the cumulated sum of normalized rewards up to the learned task, where the normalization was done for each task with the reward obtained by the task-dedicated network. Thus, the high cumulative sum corresponds to effectively combating the catastrophic forgetting (CF), and fine-tuning, which is known to suffer from CF, hovering around 1 makes sense. We observe UCL significantly outperforms EWC and different $\sigma$ values have little effect on the final reward (Re:[**R3**]). We believe the reason why EWC does not excel as in Figure 4B of the original EWC paper, [Kirkpatrick *et.al*, Overcoming catastrophic forgetting in neural networks, *PNAS* 2017], is because we consider pure continual learning setting, while the original EWC paper allows learning tasks multiple times in recurring fashion. A possible reason why UCL works so well in RL setting may be due to the by-product of our weight sampling procedure; namely, it enables effective exploration as in [20, manuscript]. We stress that there are few algorithms in the literature that work well on both SL and RL continual learning setting, and our UCL is very competitive in that sense. We will include the reward trajectories for learning each task (that resulted in Fig 1(c)) in the Supplementary Material of the final version.

**[R2]** ① We apologize for any confusions we made while describing the sampling procedure in [Line 206, manuscript]. What we meant was that we sample model parameters every iteration, and the number of sampling is 1 for each iteration. At the beginning epoch of task $t$, we sample from $q(\mathcal{W}|\boldsymbol{\theta}_t)$ with $\boldsymbol{\theta}_t = \boldsymbol{\theta}_{t-1}$ (i.e., using the learned paramter up to task $t-1$), then continue to update $\boldsymbol{\theta}_t$ in the subsequent iterations. Hopefully, this resolves the confusion of the reviewer. ② We will make sure to correct some redundant "so-called" expressions in the final version.

**[R3]** We disagree that our work is only an incremental improvement over VCL for the following reasons. ① As **[R1]** has pointed out, our novel interpretation of KL term gives new insights and variations on online Bayesian learning. ② Since UCL dramatically reduces the number of parameters compared to VCL, we can apply UCL to much larger and deeper models as shown in above experiments with CNNs. Note VCL does not have any results on using deep CNNs. ③ Since UCL samples the weight parameters only once for each iteration, applying it to actor-critic based reinforcement learning algorithm becomes possible. We believe concrete regularization terms that we derive enables such efficient sampling scheme. In contrast, VCL needs to sample weights multiple times in each iteration for the Monte-Carlo simulation, and it is almost impossible to apply VCL in the RL continual learning setting as above.

While we do not have rigorous theoretical analyses on the formulation of UCL, the combination of $\ell_1$ and $\ell_2$ norms in the regularization term reminds of the *Elastic-net*, widely used in statistical learning. Also, our node-wise notion of uncertainty gives natural extension to the CNN models by defining uncertainty for each filter (channel) and leads to good performance for deep CNN models.

[Meta-Review · NeurIPS 2019]

This paper proposed uncertainty-regularized continue learning (UCL) to address the challenge of catastrophe forgetting of neural networks. In detail, the method improves over variational continual learning (VCL) by modifying the KL regularizer in mean-field Gaussian prior/posterior setting. The approach is mainly justified by intuition explanation rather than theoretical/mathematical arguments. Experiments are performed on supervised continual learning benchmarks (split and permuted MNIST), and the method shows dominating performance over previous baselines (VCL, SI, EWC, HAT). Reviewers include experts in continual learning. Some of them were concerned on the MNIST benchmark, but with additional supervised learning and RL experiments provided in author feedback, reviewers reached a consensus of accepting this paper. Although UCL is an improvement over VCL, two continual learning experts in the reviewing panel viewed this modification as novel contribution. I would suggest the following revision In the camera ready: 1. A clear discussion on the novelty of UCL over VCL; 2. A better justification of the UCL objective, preferably with some detailed derivation; 3. Adding in the RL experiments to make the paper stronger.